# Cutaneous Polymeric-Micelles-Based Hydrogel Containing *Origanum vulgare* L. Essential Oil: In Vitro Release and Permeation, Angiogenesis, and Safety Profile In Ovo

**DOI:** 10.3390/ph16070940

**Published:** 2023-06-29

**Authors:** Ștefana Avram, Larisa Bora, Lavinia Lia Vlaia, Ana Maria Muț, Gheorghe-Emilian Olteanu, Ioana Olariu, Ioana Zinuca Magyari-Pavel, Daliana Minda, Zorița Diaconeasa, Paula Sfirloaga, Mohd Adnan, Cristina Adriana Dehelean, Corina Danciu

**Affiliations:** 1Department of Pharmacognosy, Victor Babes University of Medicine and Pharmacy, Eftimie Murgu Square, No. 2, 300041 Timisoara, Romanialarisa.bora@umft.ro (L.B.);; 2Research Center for Pharmaco-Toxicological Evaluation, Victor Babes University of Medicine and Pharmacy, Eftimie Murgu Square, No. 2, 300041 Timisoara, Romania; 3Department II—Pharmaceutical Technology, Formulation and Technology of Drugs Research Center, Victor Babes University of Medicine and Pharmacy, Eftimie Murgu Square, No. 2, 300041 Timisoara, Romania; 4Center for Research and Innovation in Personalized Medicine of Respiratory Diseases, Victor Babes University of Medicine and Pharmacy, Eftimie Murgu Square, No. 2, 300041 Timisoara, Romania; 5Department of Toxicology and Drug Industry, Victor Babes University of Medicine and Pharmacy, Eftimie Murgu Square, No. 2, 300041 Timisoara, Romania; 6Department of Food Science and Technology, Faculty of Food Science and Technology, University of Agricultural Science and Veterinary Medicine, Calea Manastur, 3-5, 400372 Cluj-Napoca, Romania; 7National Institute for Research and Development in Electrochemistry and Condensed Matter, 300569 Timisoara, Romania; paulasfirloaga@gmail.com; 8Department of Biology, College of Science, University of Hail, Hail P.O. Box 2440, Saudi Arabia

**Keywords:** *Origanum vulgare* var. *vulgare* essential oil, polymeric micelles, SEM, in vitro release, angiogenesis, chorioallantoic membrane

## Abstract

*Origanum vulgare* var. *vulgare* essential oil (OEO) is known as a natural product with multiple beneficial effects with application in dermatology. Oregano essential oil represents a potential natural therapeutic alternative for fibroepithelial polyps (FPs), commonly known as skin tags. Innovative formulations have been developed to improve the bioavailability and stability of essential oils. In this study, we aimed to evaluate the morphology of a polymeric-micelles-based hydrogel (OEO-PbH), the release and permeation profile of oregano essential oil, as well as to assess in vivo the potential effects on the degree of biocompatibility and the impact on angiogenesis in ovo, using a chick chorioallantoic membrane (CAM). Scanning electron microscopy (SEM) analysis indicated a regular aspect after the encapsulation process, while in vitro release studies showed a sustained release of the essential oil. None of the tested samples induced any irritation on the CAM and the limitation of the angiogenic process was noted. OEO-PbH, with a sustained release of OEO, potentially enhances the anti-angiogenic effect while being well tolerated and non-irritative by the vascularized CAM, especially on the blood vessels (BVs) in the presence of leptin treatment. This is the first evidence of in vivo antiangiogenic effects of a polymeric-micelle-loaded oregano essential oil, with further mechanistic insights for OEO-PbH formulation, involving leptin as a possible target. The findings suggest that the OEO-containing polymeric micelle hydrogel represents a potential future approach in the pathology of cutaneous FP and other angiogenesis-related conditions.

## 1. Introduction

*Origanum vulgare* L., also known as oregano, is an aromatic, widespread plant belonging to the *Lamiaceae* family. It is native to Mediterranean flora as well as to southwestern Eurasia [1]. Within the genus *Origanum*, this perennial species is characterized by increased variability and morphological diversity, comprising no less than six subspecies: subsp. *vulgare* L., subsp. *glandulosum* (Desfontaines) letswaart, subsp. *hirtum* (Link) letswaart, subsp. *gracile* (Kock) letswaart, subsp. *virens* (Hoffmannsegg et Link) letswaart, and subsp. *viride* (Boissier) Hayek [2]. Oregano is used in traditional medicine in order to treat digestive, respiratory, and cutaneous disorders; as a valuable spice in gastronomy; as an additional preservative in the food industry; and for agricultural and pest control purposes [3,4]. In terms of chemical composition, *Origanum vulgare* L. is characterized by the presence of volatile (essential oil) and non-volatile (flavonoids, phenolic acids, and tannins) compounds [5]. It is acknowledged that the numerous biological activities of *Origanum vulgare* L. are due to the volatile components of the plant represented by the essential oil. Although the essential oil (EO) composition varies depending on external factors such as the harvesting time, growth conditions, and geographical area, it is unanimously accepted that the main components of OEO are the monoterpenes carvacrol and thymol [6,7]. The pharmacological profile of OEO comprises a multitude of effects established throughout many years of intensive exploration. Antibacterial, antifungal, antiparasitic, anti-inflammatory, antioxidant, antitumoral, and antiproliferative effects characterize OEO and maintain the focus of scientific research on this aromatic medicinal plant [8,9,10,11,12,13]. Recent findings have highlighted the cosmeceutical potential of OEO as a natural alternative in a series of cutaneous diseases [14]. Due to the phenolic monoterpenes—carvacrol and thymol—OEO exhibits antioxidant activity that can play an important role in preventing solar skin aging [15,16]. Moreover, the statistics show that 1–1.5% of the population is confronted with wounds [17]. Thus, the anti-inflammatory and antimicrobial potential of OEO is increasingly being studied to deliver a natural option in wound healing or for other skin condition therapeutical approaches, such as acne [18,19,20,21].

Currently, researchers have shown a particular interest in the potential of various OEO-based cutaneous delivery systems as a complementary therapeutic approach to many skin disorders such as acne, skin infections, wounds, atopic dermatitis, psoriasis, and skin aging [22]. Therefore, innovative nanocarrier systems have been increasingly developed to enhance the skin bioavailability and therapeutic potential of OEO. We currently dispose of a wide range of novel nanocarriers, namely, liposomes, niosomes, nanoemulsions, nanospheres, nanocrystals, dendrimers, and even gold or silver nanoparticles [23]. The encapsulation of EOs enhances not only skin permeability and biological performance but also the compliance of the patient due to the more pleasant aesthetic appearance of the dermato-cosmetic product [24]. Moreover, the encapsulation process determines the prolonged exposure of EOs to the skin and sustained-release action, thereby providing an enhanced and effective biological effect [25]. Consequently, we designed a modern poloxamer-based hydrogel formulation to incorporate OEO as the active ingredient. The amphiphilic character of poloxamers is responsible for the increased ability to form micelles and to incorporate the hydrophobic components contained in OEO, enhancing, at the same time, the bioavailability and the biological efficacy of the EO [26,27].

The application of EOs as ointments, creams, dressings, and lotions in various cutaneous diseases has been practiced for centuries. The variability of EOs (including OEO, which is known to be the most variable species of the *Origanum* genus) can determine significant changes in chemical profile and, thus, an important increase in the toxicity risk. Therefore, it is mandatory to ensure the increased quality of raw materials and extraction methods, in accordance with the provisions of European Pharmacopoeia, to ensure an appropriate safety profile of the cosmetic product [28,29]. EOs of the *Lamiaceae* family have the potential to sensitize and irritate the skin, causing allergic contact dermatitis and hypersensitivity reactions [30]. However, the use of EOs in topical preparations is generally safe, efficient, and less expensive when they are used in a decreased concentration or a diluted form, and the quality of the formulation is assured [17]. Considering the safety issue for topical formulations, the poloxamers selected for our formulation are frequently used as excipients due to their biocompatible, non-toxic, and non-irritant features, which are recognized by FDA as GRAS (generally recognized as safe) and described by the European and US Pharmacopoeia. Moreover, the use of poloxamers as excipients in topical formulations can be an advantageous option also due to their multiple functionalities, acting as solubilizers for lipophilic bioactive compounds, gelling agents, and permeation enhancers [31,32].

In our previous work, we successfully showed that the polymeric hydrogel containing OEO proved to have a significant effect on decreasing the proliferation and migration of human keratinocytes [33]. These results were encouraging for potential beneficial use on cutaneous impaired cell proliferation and differentiation manifested in FPs. The pathogenic profile of the FP involves dysregulated keratinocyte and fibroblast pathways, next to a higher vascular density and abnormalities. A few studies have evaluated the potential effect of oregano essential oil on the process of angiogenesis. *Origanum onites* L. essential oil induced an anti-angiogenic effect by inhibiting migration and tube formation, using an endothelial cell assay [34]. Additionally, carvacrol, the main phytochemical in OEO, tested as a nanoformulation was shown to possess anti-angiogenic activity, reducing the expression of angiogenic factors VEGF, COX-2, and CD-31 in an animal study [35]. Considering the efforts to reduce the use of animal models, the CAM stands as an accepted in vivo alternative to assess skin irritation, an intermediate step prior to further evaluation. In this regard, the CAM assay was considered for the in vivo evaluation of the OEO formulation. This is the first study involving a polymeric-based formulation containing oregano essential oil using the CAM assay. The protocol makes use of a chorioallantoic membrane of a chick embryo, commonly used to perform angiogenic studies, in cancer research as an in vivo alternative to animal studies, with several advantages such as costs, time, and ease of access [36]. The hen egg test–chorioallantoic membrane (HET-CAM) assay has also gained attention in assessing irritability in the field of bioengineering, ophthalmic products as well as cosmeceuticals, as an alternative to the Draize eye rabbit test, according to the 3Rs principles: replacement, reduction, and refinement. [37,38,39,40].

Therefore, in the current study, we considered the evaluation of the same innovative OEO-containing polymeric micelles formulation in terms of cutaneous application. One of the aims was the assessment of the releasing pattern of the formulation. The second purpose was to study the in vivo tolerability of the OEO-PbH formulation and the potential effect on the angiogenic process in ovo on the CAM assay, as a step forward toward a future clinical evaluation on FP.

## 2. Results

### 2.1. SEM Analysis

The surface morphologies of the polymeric-micelles-based hydrogel containing OEO, next to OEO alone, and a blank hydrogel were investigated using SEM. As depicted in Figure 1, the micron particles incorporated in the polymer matrix have a regular aspect after the encapsulation process, as indicated by a mostly spherical smooth shape. Essential differences between the surface morphology of OEO and OEO-PbH were observed.

### 2.2. In Vitro Drug Release and Skin Permeation Studies

#### 2.2.1. In Vitro Release of OEO

The in vitro release profile of OEO from the poloxamer-binary hydrogel through the polysulfone membrane is shown in Figure 2a, and the values of the process-specific parameters are listed in Table 1.

It can be observed that the polymeric gel vehicle provided a sustained release of OEO throughout the test period without reaching a steady state, with a total released amount of 33.33 ± 4.28% OEO, corresponding to 2702.34 ± 5.12 μg/cm^2^ of the initial content. Additionally, the release profile of OEO presented a lag time (the x-axis intercepts at the linear segment of the curve for permeated cumulative OEO amount vs. time, under steady-state conditions), with a value of less than 30 min, accepted as a common characteristic for this profile (Figure 2a and Table 1).

The data of in vitro OEO release test fitted best with the zero-order model, for which a higher determination coefficient value (R^2^ = 0.9962) was obtained (Table 2).

#### 2.2.2. In Vitro Pig Ear Skin Permeation of OEO

The specific permeation profile of OEO through excised pig ear skin depicted in Figure 2b revealed a prolonged release process leading to a final percentage of OEO released of 23.58%, corresponding to 1915.442 μg/cm^2^ of the initial content in experimental formulation (Figure 2b). Furthermore, over a 24 h period, this profile was biphasic, presenting an initial faster phase of 8 h length and a second phase, which was slower and longer (of 14 h length) than the first phase (Figure 2b). The steady-state flux value calculated for the initial phase was 3.12-fold higher than that calculated for the second phase (Table 1).

In the first phase of the transcutaneous permeation of OEO, there was also revealed a significant lag time (1.65 ± 2.38 h), but even so, in the first 8 h of testing, the polymeric-micelles-based hydrogel produced a faster and higher transfer of OEO.

Similar to the in vitro release study, the data obtained from the in vitro OEO permeation from OEO-PbH formulation through pig ear skin were fitted to the same mathematical models, and the results of the kinetic analysis are listed in Table 2. Based on the highest value of the determination coefficient (R^2^ = 0.9531), it can be suggested that the Korsmeyer–Peppas model best described the in vitro skin permeation of OEO from the tested formulation (Table 2), demonstrating the involvement of several phenomena in this process. Further, the calculated value of the diffusion exponent, *n*, (indicative of the release mechanism of the active ingredient) was higher than 0.89 (Table 2), indicating a super case-II drug transport mechanism.

### 2.3. OEO-PbH Biocompatibility In Vivo on the CAM

To determine the degree of biocompatibility of OEO-PbH, we used an in ovo HET-CAM assay. The method is intended to assess the irritative potential of the samples with applicability for dermal and mucosal tissues.

OEO-PbH did not induce any changes in the vascularized CAM, with no signs of hemorrhage, coagulability, or vascular lysis observed either during the 5 min of observation (Figure 3) or 24 h after the administration, compared to the effect of sodium lauryl sulfate (SLS), used as the positive control, that induced irritation and hemorrhage on the membranes. The occurrence of an irritative reaction was not observed either for the blank hydrogel or for the pure oregano essential oil, as indicated by the calculated irritation scores (Table 3). There was no sign of embryotoxicity detected throughout the experiment.

### 2.4. OEO-PbH Modulates Angiogenesis In Ovo

Using the CAM assay, we investigated in ovo the effect of OEO-PbH formulation on the process of angiogenesis, as well as the impact on the involved tissues, aiming to obtain in vivo evidence of OEO formulation on cutaneous highly vascularized tissues.

The samples used to treat the CAMs were in a concentration of 200 µg/mL, and the effect induced was monitored using stereomicroscopic observation after 24 and 48 h, during a period of rapid growth of the vascular network. OEO-PbH induced an effect on the capillaries in formation, showing a thin capillary network, with a reduced degree of interconnection in comparison with Blank-PbH, where the BVs in formation displayed a more dilated aspect and a higher branching pattern. CAMs treated with OEO alone showed a reduced number of capillaries, but with a relatively dilated aspect. The stereomicroscopic images express the reduced number of newly formed capillaries in the case of OEO and OEO-PbH, especially 48 h after treatment (Figure 4a). Samples containing OEO were devoid of toxicity at vascular and CAM levels. Supporting the stereomicroscopic observation, the semi-quantitative morphometric analysis showed OEO-PbH and OEO expressed similar effects, decreasing the number of BVs around the application spot, with a calculated angiogenic inhibition index significantly higher compared to Blank-PbH and the control (Figure 4b).

### 2.5. OEO-PbH Modulates Leptin-Induced Angiogenesis In Ovo on the CAM

We investigated the possible impact of the studied formulation on the angiogenesis process stimulated by leptin, an overexpressed factor in pathologies such as cutaneous FPs. Consequently, we evaluated the changes induced in the chorioallantoic membrane by leptin in co-administration with the OEO samples.

In the case of samples containing OEO, a reduced degree of vascularization was noted when compared with leptin alone (Figure 5a). Blank-PbH administration did not influence the vessel density around the application spot, but an injurious phenomenon was recorded, which could not be observed in the case of membranes treated with samples containing OEO. OEO-PbH induced a lower number of small-caliber vessels, showing a more intense limitative effect on the leptin angiogenic response (Figure 5b).

### 2.6. Histological Observations

Histological analysis of the CAM provided valuable insights into the effects of OEO and OEO-PbH on tissue architecture and angiogenesis.

Hematoxylin and eosin (H&E) stained sections of the CAM revealed distinct changes in the allantoic ectoderm (AE), BVs, and mesodermal layer (ML) following treatment with Blank-PbH, OEO, and OEO-PbH (Figure 6). The presence of the chorionic endoderm (CE) was also observed in the CAM treated with OEO and OEO-PbH, indicating a potential effect of these treatments on the differentiation of the CAM.

In the control CAM, the AE, BVs, ML, and fibroblasts were clearly identifiable, providing a baseline for comparison. Treatment with Blank-PbH did not appear to significantly affect the BVs, while the ML exhibited signs of edema.

Treatment with OEO resulted in a reduced ML, thinned AE, and a decrease in the number of BVs. While after treatment with OEO-PbH, ML underwent a complete architectural change, the AE was thinned, and the BVs were reduced in number and size.

## 3. Discussion

Considering that the efficacy of a topical pharmaceutical preparation depends on the release of the bioactive component, the in vitro release profile constitutes an important parameter for product quality control. The in vitro release test of the active component from the semisolid preparation assesses the product performance, namely, those attributes that correlate with the in vivo performance of the active ingredient [41]. It is generally accepted that the release of the active component from semisolid preparations depends significantly on several factors, including its solubility in the vehicle, and also the type and viscosity of the carrier [42].

In the present in vitro release study, the OEO delivery carrier was a micellar hydrogel based on two poloxamers in order to solubilize (encapsulate) the lipophilic active component in the hydrophobic core of the polymeric micelles dispersed in the aqueous phase. Moreover, in this micellar gel system, the mobility and deformability of the micelles is strongly restricted [43,44].

Therefore, the encapsulated active component was slowly released by diffusion through the polymeric gel matrix. Sustained topical drug delivery from poloxamer-based hydrogels was described by other previously published studies, which also suggest that hydrogel viscosity and the partition of the active component between the micelles and aqueous phase are the main factors affecting this type of release [45,46]. The final percentage (33.33 ± 4.28%) of OEO released from the tested formulation was much lower than that recommended as “ideal” by the EMA draft guideline [47], but the other requirements for the in vitro release test were accomplished. The in vitro OEO permeation through pig ear skin was also a prolonged release process, which can be attributed to the above-discussed mentions.

Various polymeric carriers are currently being researched for the encapsulation of essential oils to enhance their properties and biological benefits [48]. Such innovative formulations are intensively investigated in pathologic conditions involving a disrupted angiogenesis process because they can improve drug–tissue diffusion, with a low risk of systemic toxicity [49].

Oregano essential oil, as a potent bioactive source of active compounds, with multiple pharmacological and biotechnological benefits [14], was submitted to encapsulation designs and delivery, and multiple activities have been reported by others [50,51].

Our polymeric micelle formulation, showing a sustained, prolonged release profile, was submitted to an in ovo investigation, and the lack of irritability and good bioavailability was proven using the HET-CAM assay, an optimal alternative in vivo test for toxicity studies [52].

Using the in ovo CAM assay, we also conducted an investigation on the potential impact of the novel formulation on the angiogenic process. Studies regarding the effect of oregano essential oil have previously been performed, but these were mainly in vitro and employed different species, such as *Origanum onites* L. [34]. Others explored in vitro and animal studies, the main active component of oregano essential oil, carvacrol, in an encapsulated formulation, showing the potential inhibitory effect on angiogenesis by downregulating stimulating factors such as COX-2, VEGF, and CD31 [35].

Less data are available on the impact of *Origanum vulgare* essential oil on dysregulated angiogenic pathways in skin-related diseases. A study proved the anti-inflammatory and immunomodulatory activities of the essential oil of *O. vulgare* in a skin model [53]. Our evaluation contributes by adding important data regarding the effects of *O. vulgare* essential oil encapsulated in a polymeric micelle formulation on the dysregulated process of angiogenesis. OEO-PbH induced a reduced number of BVs in the developing CAM while impacting the CAM tissues. Furthermore, an important effect was obtained by OEO-PbH formulation in reducing the number of BVs in leptin-pretreated CAMs, suggesting a possible implication of the prolonged-release OEO in the leptin-activated angiogenic pathways.

The histological observations of the CAMs following treatment with OEO and OEO-PbH further elucidated the effects on angiogenic CAM tissues.

Changes in the AE, BV, and ML following treatment with Blank-PbH, OEO, and OEO-PbH were observed. As revealed by the presence of CE on the treated CAMs, OEO and OEO-PbH display a potential effect on the differentiation of the CAM.

Without affecting the BVs architecture, treatment with Blank-PbH exhibited signs of edema in the ML, suggesting a change in the permeability of the BVs. This could potentially be attributed to the physical properties of the hydrogel, which may have affected the fluid balance in the tissue.

OEO induced a lower number of BVs in the CAM, suggesting a potential anti-angiogenic effect of OEO. The presence of CE in the OEO-treated CAM further supports this observation, as CE is known to play a role in the regulation of angiogenesis [54].

The most significant changes were observed in the CAM treated with OEO-PbH, conducting changes in ML and AE while leading to a reduced number of smaller BVs. These findings suggest that the encapsulation of OEO in the polymeric-micelles-based hydrogel enhanced its anti-angiogenic effects, potentially by improving the bioavailability and stability of OEO.

The histological findings align with the results of the in ovo HET-CAM assay, which showed that OEO-PbH did not induce any irritation or hemorrhage on the CAM, indicating a high degree of biocompatibility.

Furthermore, SEM analysis revealed a regular aspect of the micron particles in the polymer matrix after the encapsulation process, suggesting that the formulation process did not adversely affect the physical properties of the hydrogel. A prolonged release of OEO from the polymeric-micelles-based formulation was observed in vitro, using pig ear skin, with a faster release for the first 8 h, followed by a slower process of release. This pattern of the liberation of OEO from polymeric micelles suggests the potential beneficial effect in limiting the number of BVs in the ML, next to modifications in the ML and AE of the CAM, without causing toxic events.

Applying an in ovo protocol using the chorioallantoic membrane of the avian embryo, we have established a lack of irritability induced by OEO-PbH, hence suggesting the biocompatibility and safety characteristics for topical administration. Further, the CAM assay provided insight into the effect of OEO-PbH on limiting the number of BVs in formation. A more intense effect was induced by this formulation on the BVs in the presence of leptin treatment, an overexpressed adipokine in the pathology of FPs, thus being indicative of a possible mechanism of action. The histological analysis provided valuable insights into the effects of OEO and OEO-PbH on tissue architecture and angiogenesis in CAM. The findings suggest that OEO-PbH has potential as a non-irritating, biocompatible, and anti-angiogenic formulation on the chorioallantoic membrane, a valuable experimental setting as a preliminary evaluation before cutaneous application. Our results, showing the effect of OEO-PbH on reducing the number of newly forming vessels, and the limitation of leptin-induced angiogenesis represent relevant findings for the possible treatment of FP, commonly known as skin tags. Further studies, including clinical evaluations, are needed, and represent the important perspectives of the present study, to confirm these findings and to explore the potential therapeutic applications of OEO-PbH in cutaneous applications.

Therefore, the essential oil of oregano, and especially the formulation based on the polymeric micelles containing OEO, can represent a safe therapeutic alternative, biocompatible with topical administration, while exerting limiting effects on the angiogenic phenomenon activated in the pathology of cutaneous fibroepithelial polyps.

## 4. Materials and Methods

### 4.1. Materials

#### 4.1.1. Plant Materials

The essential oil was obtained from aerial parts of *Origanum vulgare* var. *vulgare*, collected from the western part of Romania and identified in the Department of Pharmacognosy, University of Medicine and Pharmacy “Victor Babes” Timișoara (specimen number: OV5/2020). The extraction was carried out by hydrodistillation, as described by Bora et al. [33].

#### 4.1.2. Chemicals

To prepare the poloxamer binary hydrogels, we used Poloxamer 407 (Pluronic F127, CAS no.: 9003-11-6, product number: P2443) and Pluronic L 31 (product number: 435406), acquired from Sigma-Aldrich-Chemie (Steinheim, Germany). For the chorioallantoic membrane assay, sodium dodecyl sulfate (SDS, CAS no.: 151-21-3; MW 288.38 g/mol) and leptin from mice (L3772, CAS no.: 181030-10-4; predicted MW~16 kDa) were purchased from Sigma-Aldrich-Chemie (Steinheim, Germany). All used reagents were analytical-grade.

### 4.2. Preparation of Polymeric-Micelles-Based Hydrogels

Two polymeric-micelles-based hydrogels (blank and OEO-containing formulation) were prepared using a “cold process” [55], as detailed in our previous study [33]. Blank poloxamers-based hydrogel was prepared by dispersing Pluronic F127 (P127) and Pluronic L 31 (P31) in concentrations of 20% *w/w* and 1% *w/w*, respectively, in cool (4 °C) purified water. The dispersion was kept for a minimum of 24 h at 4 °C while stirring to obtain a transparent, homogeneous solution, indicating total dissolution of the poloxamers. OEO at 5% *w/w* concentration was uniformly dispersed in the cold fluid poloxamers-based vehicle, previously prepared by the same method as the blank formulation. Stirring was continued at room temperature until a transparent, homogeneous hydrogel was obtained.

### 4.3. SEM

The surface morphology of the studied samples was investigated using SEM, an advanced method widely used in order to characterize ultra-morphological and structural particularities of modern formulations, such as size, shape, and structure, at high resolution, thus evaluating the formation of nanoparticles, aggregates, or micelles [56,57]. Surface morphologies of the OEO-developed hydrogels were scanned and analyzed using photomicrographs. In order to obtain a good resolution of the images, the samples were analyzed using SEM (Inspect S), FEI Company Netherlands, in low-vacuum mode using an LFD detector, spot value of 2.5, and high voltage (HV) of 25 kV.

### 4.4. OEO In Vitro Release and Skin Permeation Studies

For the experiments of in vitro release and skin permeation of OEO from poloxamer-binary hydrogel, quantitative in vitro release/permeation tests were conducted in accordance with the recommendations of the regulatory authorities describing these assays [47,58]. In these tests, one of the most widely used systems, namely, static vertical diffusion cell system, was used. Additionally, the experimental conditions (the choice of the membrane and receptor medium, the sampling time, and the analytical method) were set up in accordance with the regulatory guidance in force [47,58,59]. Therefore, a system of six Franz diffusion cells (Microette-Hanson, model 57-6AS9, Chatsworth, CA, USA) was used. Each vertical diffusion cell has an effective diffusional area of 1.767 cm^2^ and an acceptor compartment volume of 6.5 mL. The used receptor medium was phosphate saline buffer (pH, 7.4) with 30% (*v*/*v*) ethanol (Merck, Darmstadt, Germany; 99.8% purity; CAS no.: 64-17-5; MW 46.07 g/mol), which fulfilled the sink conditions for OEO during the in vitro release and skin permeation tests. Each in vitro release and skin permeation experiment were performed five times.

#### 4.4.1. In Vitro OEO Release Studies

The in vitro OEO release measurements were performed according to the method described by USP/NF [58]. Practically, the acceptor compartments were filled with the freshly prepared and deaerated receptor medium and then covered by the synthetic hydrophilic polysulfone membranes (0.45 μm Supor^®^ membranes, Pall Corporation, Port Washington, NY, USA), which were previously soaked in the receptor medium for 30 min. The poloxamer-binary hydrogel containing OEO (approx. 0.300 g) was weighed in the donor compartment, which was fixed on the receptor cell, facing the synthetic membrane and avoiding air bubble incorporation.

During the experiments, the diffusion cells system was maintained at 32 ± 2 °C and the receptor medium was constantly stirred at 600 rpm. Samples of 0.5 mL were automatically withdrawn from the receptor medium at 0.5, 1, 2, 3, 4, 5, 6, and 7 h, and replaced with equal volume of fresh receptor fluid to provide sink conditions.

The concentration of OEO in the collected samples was analyzed using UV spectrophotometric method (UV-VIS T70 plus spectrophotometer with UVWIN5 software, PG Instruments, Lutterworth, UK) at a wavelength of 273 nm, at which OEO has maximum absorption in the selected receptor medium (phosphate saline buffer solution with 30% ethanol). The assay was linear in an OEO concentration range of 14.112–141.12 μg/mL (y = 0.1714x + 0.061, R^2^ = 0.9996).

#### 4.4.2. In Vitro OEO Skin Permeation Studies

For the in vitro OEO skin permeation study, dermatomed ear skins from young pigs were used as model membranes. Fresh-cut pig ears from 4-month-old animals were cleaned with tap water and shaved. From the outer part of the pig ears, skin strips with a 500 μm thickness were removed using dermatome (Acculan 3 Ti Electrodermatome, Aesculap—a BBraun Company, Hazelwood, MO, USA). If the dermatomed skin was not immediately used for the in vitro permeation test, it was stored at −18 °C for up to 2 months and, before use, was thawed at room temperature. Squares of 2 cm^2^ were cut from the excised intact skin, without damage.

In vitro skin permeation test was performed following the method described above, only that the sample aliquots were withdrawn at intervals of 1, 2, 3, 4, 5, 6, 7, 8, 9, 10, 11, 12, 13, 14, 15, 16, 17, 18, 20, 22, and 24 h. OEO quantitative determination from the collected samples was performed by the same UV spectrophotometric assay.

#### 4.4.3. Data Analysis of In Vitro OEO Release and Permeation Studies

The cumulative amount of OEO permeated per membrane unit area (μg/cm^2^) into the receptor medium was plotted versus time (t, h) to calculate the steady-state flux (J_SS_, μg/cm^2^/h), the permeability coefficient (K_p_, cm/h), and the lag time (t_L_, h).

To evaluate the in vitro OEO release and skin permeation processes, the respective experimental data were fitted to four currently used kinetic models for semisolids [60,61,62]:Zero-order model: M_t_ = M_0_ + K_0_t,(1)
where M_t_ is the amount of drug dissolved during time, t; M_0_ is the initial amount of drug in solution (usually zero); and K_0_ is the zero-order release constant expressed in units of concentration/time. This model indicates that the active ingredient was released at a constant rate from the delivery system (i.e., the release rate did not depend on the amount of activity that remained in the delivery system).
First-order model: logC = logC_0_ − K_1_t/2.303,(2)
where C_0_ is the initial concentration of drug, K_1_ is the first-order release constant, and t is the time. First-order model suggests that during dissolution, the active compound molecules diffuse through the gel-like layer, which is formed around it.
Higuchi model: M = K_H_ t^1/2^, (3)
where M is the amount of active substance released at time, t, and K_H_ is the Higuchi release constant. This model points out that the release rate is dependent on active molecule diffusion through polymer matrix of the hydrogel; hence, the release rate depends on the polymeric micelle size.
Korsmeyer–Peppas model: M_t_/M_∞_ = K_P_t^n^, (4)
where M_t_/M_∞_ is the ratio of the amount of substance released at time, t; K_P_ is the Korsmeyer–Peppas release rate constant; and n is the diffusion coefficient, describing the release mechanism of the active ingredient (quasi-Fickian diffusion for n < 0.5, diffusion for n = 0.5, non-Fickian diffusion for 0.5 < n < 1, case II transport and super case II transport for n = 1 respectively n > 1). The Korsmeyer–Peppas model is a comprehensive semi-empirical model that can generally describe diffusion in a controlled matrix system.

To assess the fit quality of each mathematical model, linear regression analysis was used. The kinetic model leading to the highest value of the determination coefficient (R^2^) was considered the most convenient to describe the release of OEO from the experimental poloxamer-binary hydrogel.

### 4.5. In Ovo CAM Assay

#### 4.5.1. CAM Assay

The CAM assay is considered an in vivo technique that allows the investigation of effects induced by the tested samples upon the developing, intensively vascularized CAM. Briefly, the method makes use of fertilized hen (*Gallus gallus domesticus*) eggs and involves egg cleansing with 70% ethanol before incubation at controlled 37 °C and 50% humidity. On the third embryonic day of development (EDD 3), 3–4 mL of albumen were removed, and an opening was cut on the upper side of the eggs on EDD 4 [63]. Macroscopic evaluation was performed daily in ovo by means of stereomicroscopy (ZEISS SteREO Discovery.V8, Göttingen, Germany), and all images were registered and processed by Axiocam 105 color, AxioVision SE64. Rel. 4.9.1 Software, (ZEISS Göttingen, Germany), ImageJ (ImageJ Version 1.50e, https://imagej.nih.gov/ij/index.html, accessed on 26 May 2023), and GIMP software (GIMP v 2.8, https://www.gimp.org, accessed on 26 May 2023).

#### 4.5.2. Irritability Assessment by the HET-CAM Protocol

The samples were evaluated in order to assess the potential irritability on cutaneous tissues using a modified protocol of the HET-CAM test, which is known as an alternative protocol for evaluating the potential irritative effect of compounds intended for ophthalmic and dermatologic products, or as a method of assessing in vivo biocompatibility [64,65,66]. The protocol used here was an adapted alternative to the HET-CAM method, as recommended by ICCVAM guidelines [67].

The fertilized eggs prepared as described above were incubated until the 9th day of incubation. At this point, by stereomicroscope assistance, 0.3 mL of the samples were applied onto the membrane. The positive control was SDS 1%, while the negative control was represented by distillate water. The tested membranes were monitored over a period of 300 s, by means of a stereomicroscope, while inspecting for irritative phenomena; the time for the occurrence of the selected irritation parameters (hemorrhage, *H*; vascular lysis, *L*; coagulation, *C*) was recorded in seconds. The irritation score (*IS*) was then calculated using the following equation:$$IS=5\times \left[\frac{301-Sec\;H}{300}\right]+7\times \left[\frac{301-Sec\;L\;}{300}\right]+9\times \left[\frac{301-Sec\;C}{300}\right]$$
where hemorrhage (*Sec H*) is the start of observation (in seconds) of bleeding reactions on the membrane, lysis time (*Sec L*) is the start of observation (in seconds) of lysis of the vessel on the membrane, coagulation time (*Sec L*) is the start of observation (in seconds) of the formation of coagulation on the membrane. Irritation scores were then compared to a grading system established by Luepke [68]. Experiments were carried out in triplicate.

#### 4.5.3. Angiogenesis Effects Using the CAM Assay

In order to detect possible effects of the samples on the rapidly developing blood vessels, the angiogenesis in ovo CAM assay was performed. From day 0, 0 h (EDD 7) samples were applied daily, the specimens were visually inspected and relevant photographs were taken by means of a digital camera attached to a stereomicroscope (ZEISS SteREO Discovery.V8, Göttingen, Germany), and all images were registered by Axiocam 105 color (ZEISS Göttingen, Germany) and processed using AxioVision SE64. Rel. 4.9.1 Software, (ZEISS Göttingen, Germany), ImageJ (ImageJ Version 1.50e, https://imagej.nih.gov/ij/index.html, accessed on 26 May 2023), and GIMP software (GIMP v 2.8, https://www.gimp.org, accessed on 26 May 2023). Volumes of 5 µL/egg of all samples were applied directly inside plastic rings (3 mm in diameter) previously placed on top of the CAM. In the first experimental assessment, the angiogenic response was monitored from EDD 7 for the following group of samples: (1) Control, Blank-PbH; (2) OEO; (3) OEO-PbH.

Secondly, we intended to evaluate the effect of OEO samples in a microenvironment characterized by higher levels of leptin, an adipokine with pleiotropic functionality, stimulating keratinocyte proliferation and angiogenesis. Hence, we investigated the angiogenic response after exposure to the same OEO samples upon leptin (100 µg/mL)-pretreated chorioallantoic membranes, using the following experimental groups: (1) leptin control (Lep); (2) Blank-PbH + Lep; (3) OEO + Lep; (4) OEO-PbH + Lep.

For both angiogenesis experiments, the response was daily inspected for the treated specimens using stereomicroscopy, and relevant images were submitted to a semi-quantitative morphometric evaluation of the impact on the angiogenic process. As proposed by others [69,70,71], we employed counting the number of blood vessels (*BVs*) intersecting the inoculation ring. In order to express a score representing angiogenesis inhibition (*AI*) in percentages, the following equation was used:$$AI\;\left(\%\right)=\left[1-\frac{No\;BVs\;test}{No\;BVs\;blank}\right]\times 100$$

### 4.6. Histoprocessing and Histology Evaluation

#### 4.6.1. Tissue Collection and Processing

On the first day of CAM incubation, the treated areas of the CAM were excised and immediately fixed in 10% neutral buffered formalin (Epredia, Kalamazoo, MI, USA) for 24 h. The tissues were then dehydrated in a graded series of ethanol, cleared in xylene (Epredia, Kalamazoo, MI, USA), and embedded in paraffin wax (Epredia, Kalamazoo, MI, USA). Sections of 5 µm thickness were cut using a microtome and mounted on glass slides.

#### 4.6.2. H&E Staining

The tissue sections were deparaffinized in xylene and rehydrated through a graded series of ethanol to distilled water. The sections were then stained with hematoxylin (Epredia, Kalamazoo, MI, USA), differentiated in acid alcohol (Epredia, Kalamazoo, MI, USA), and blued in running tap water. After rinsing, the sections were counterstained with eosin (Epredia, Kalamazoo, MI, USA), dehydrated, cleared, and mounted with a coverslip using a resinous medium (Epredia, Kalamazoo, MI, USA).

#### 4.6.3. Microscopic Analysis

The stained sections were examined under a light microscope, and photomicrographs were taken using a Leica ICC50W camera (Leica Microsystems, Wetzlar, Germany) mounted on a Leica DM750 microscope (Leica Microsystems, Wetzlar, Germany). The AE, BV, ML, and fibroblasts were identified in the control CAM. Changes in the CAM architecture, including the ML, AE, and BVs, were noted after treatment with Blank-PbH, OEO, and OEO-PbH. The presence of CE was also observed in the CAM treated with OEO and OEO-PbH.

The degree of angiogenesis, the number and size of BVs, and the thickness of the AE and ML were evaluated and compared between the control and treated CAM. The potential irritation and edema in the ML were also assessed.

### 4.7. Statistical Analysis

Statistical analysis was performed with GraphPad Prism 8 software (San Diego, CA, USA). For the angiogenesis results, the number of BVs intersecting the ring was counted, calculating the Angiogenic Index in relation to control specimens. Comparison among the treated and untreated groups was performed using the one-way ANOVA followed by Dunnett’s multiple comparison test (* *p* < 0.05; ** *p* < 0.01; *** *p* < 0.001). Data were represented as mean ± SD.

## 5. Conclusions

A simple and biocompatible OEO poloxamer binary hydrogel was successfully formulated and prepared. The OEO-PbH formulation produced a sustained, prolonged release of OEO through the hydrophilic membrane and pig ear skin.

Applying the in ovo method of the chorioallantoic membrane, an exceptional alternative model that complies with the 3R principle of replacement, reduction, and refinement, we established the lack of irritability induced by OEO-PbH and identified an effect of limiting the number of capillaries, especially on BVs, in the presence of leptin treatment, with applicability in the pathology of cutaneous FP.

The incorporation of adipokine-like leptin in the study design also offers insights into the formulation’s potential role in pathologies associated with abnormal angiogenesis and offers a novel avenue for future research. Future studies could investigate the interaction between such adipokines and this micellar hydrogel formulation in more detail, expanding the understanding of its therapeutic potential in FP and perhaps other conditions associated with abnormal angiogenesis.

Significantly, this research provides a potential foundation for further exploration into the clinical utility of OEO-PbH in the management of cutaneous fibroepithelial polyps. The demonstration of OEO-PbH’s biocompatibility, lack of irritability, and limiting effect on angiogenesis suggests the potential for its therapeutic application, and its sustained release could enhance patient compliance and improve therapeutic outcomes.

Moreover, the findings also shed light on the possible action mechanism of the formulation, with the sustained release of OEO potentially enhancing its anti-angiogenic effects. Finally, this is the first evidence of the in vivo antiangiogenic effects of a polymeric-micelle-loaded oregano essential oil. Next to the lack of irritation on CAM tissues, the OEO-PbH formulation was proven to have good bioavailability and safe use intended for cutaneous applications. Moreover, the results point to mechanistic insights into the OEO-PbH formulation, involving leptin as a possible target. However, further research is necessary to substantiate these findings, investigate the exact mechanism of action, and optimize the formulation for maximal therapeutic efficiency and patient comfort in clinical settings.

## Figures and Tables

**Figure 1 pharmaceuticals-16-00940-f001:**
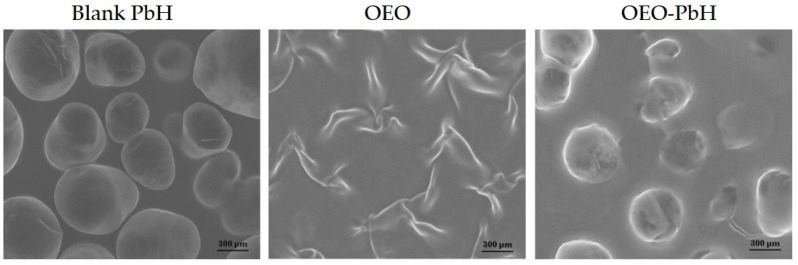
SEM images of polymeric-micelles-based hydrogel (Blank-PbH), OEO, and polymeric-micelles-based hydrogel containing OEO (OEO-PbH).

**Figure 2 pharmaceuticals-16-00940-f002:**
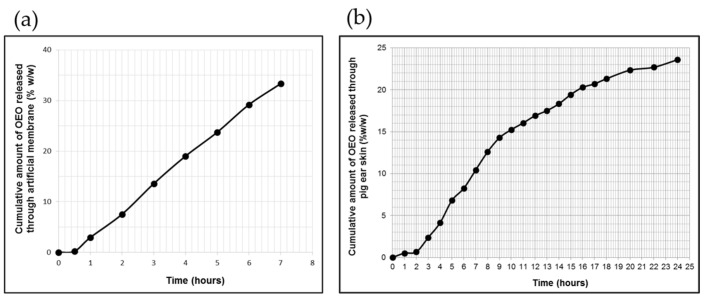
The profiles of cumulative OEO release (**a**) and skin permeation (**b**) from OEO-PbH versus time (mean ± SD of five replicates, *p* < 0.05).

**Figure 3 pharmaceuticals-16-00940-f003:**
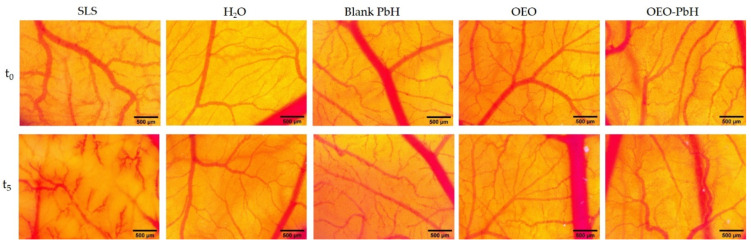
The in ovo stereomicroscopic images of the HET-CAM assay illustrating effects initially (t_0_) and 300 s (t_5_) after contact with the samples: irritation and hemorrhagic aspects of the CAM after SLS treatment; no irritation was noted in the case of Blank-PbH and H_2_O distillate; the lack of irritability by the application of the samples OEO and OEO-PbH; scales bars represent 500 µm.

**Figure 4 pharmaceuticals-16-00940-f004:**
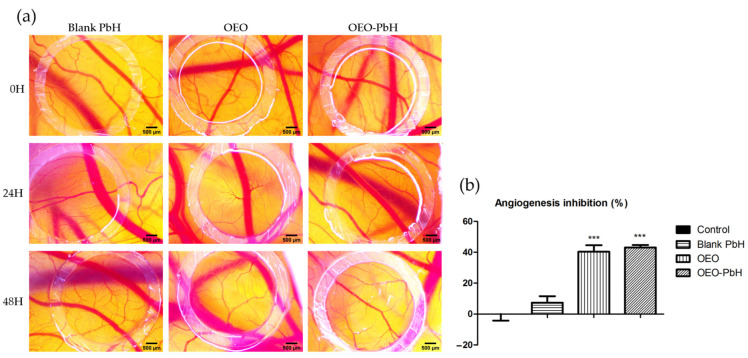
Effects of OEO-PbH on normal angiogenesis on the CAM. Stereomicroscopic images of OEO-PbH-induced effects compared to OEO and Blank-PbH (**a**) and the angiogenic inhibition index (%) (**b**). Data are expressed as mean ± SD. One-way ANOVA and Dunnett’s multiple comparison post-test were used for comparison among groups (*** *p* < 0.001 vs. Control).

**Figure 5 pharmaceuticals-16-00940-f005:**
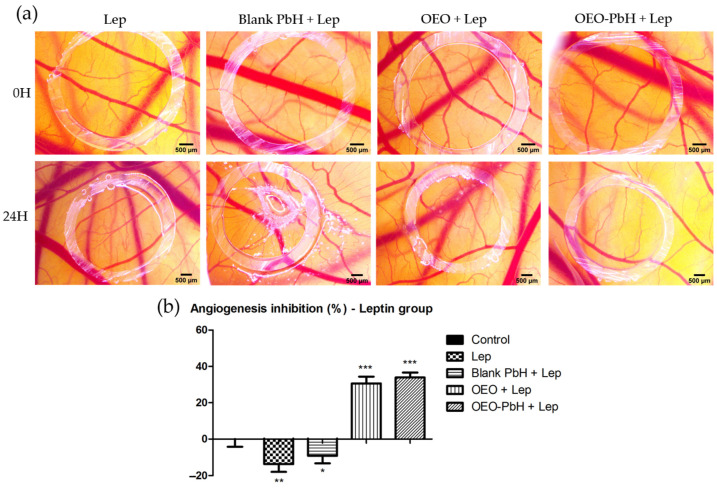
Effects of OEO-PbH on BVs for the CAM after application of leptin. Stereomicroscopic images of OEO-PbHinduced effects compared to OEO and Blank-PbH (**a**) and the angiogenic inhibition index (%) calculated in the case of leptin coadministration (**b**). Data are expressed as mean ± SD. One-way ANOVA and Dunnett’s multiple comparison post-test were used for comparison among groups (* *p* < 0.05, ** *p* < 0.01, *** *p* < 0.001 vs. Control).

**Figure 6 pharmaceuticals-16-00940-f006:**
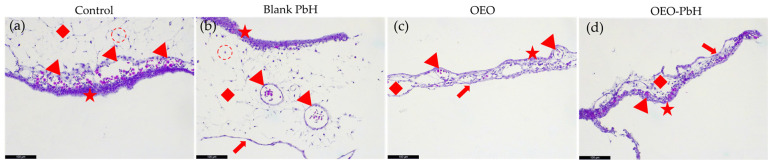
Microscopic analysis of CAM sections using H&E. (**a**) Control CAM incubation day 1, red star—AE, red arrowhead—BV, red diamond—ML, red dotted circle—fibroblasts; (**b**) CAM incubation day 1 treated with Blank-PbH, while no clear effect on BVs is noted, the ML appears edematous signifying a change in the permeability of the BVs; (**c**) CAM incubation day 1 treated with OEO, reduced ML, thinned AE, BVs are still visible but reduced in number, red arrow—CE; (**d**) CAM incubation day 1 treated with OEO-PbH, complete architectural change in ML, thinned AE, BVs are reduced in number and size.

**Table 1 pharmaceuticals-16-00940-t001:** Specific parameters of OEO release and permeation from polymeric-micelles-based hydrogel through synthetic membrane and pig ear skin.

Membranes Used in In Vitro OEO Release/Permeation Study	J_SS_ (μg/cm^2^/h)	K_P_ × 10^−6^ (cm/h)	t_L_ (h)
Synthetic polysulfone membrane	419.99 ± 8.12	83.99 ± 1.32	0.44 ± 1.47
Pig ear skin	155.06 ± 6.57 (2–10 h)	31.01 ± 4.56	1.65 ± 2.38
	49.63 ± 3.19 (10–24 h)	9.92 ± 3.05	–

**Table 2 pharmaceuticals-16-00940-t002:** Results of kinetic analysis for OEO in vitro permeability data, from the polymeric-micelles-based hydrogel through synthetic membrane and pig ear skin.

Membranes Used in In Vitro OEO Release/ Permeation Study	Zero Order		First Order		Higuchi		Korsmeyer–Peppas		
	K_0_ (μg/h)	R^2^	K_1_ (h^−1^)	R^2^	K_H_ (h^−0.5^)	R^2^	K_P_ (h^−n^)	n	R^2^
Synthetic polysulfone membrane	5.0374	0.9962	0.0604	0.9932	10.1080	0.8222	0.1831	1.7605	0.9322
Pig ear skin	1.1015	0.9278	0.0126	0.9425	4.6450	0.8652	0.3346	1.4833	0.9531

**Table 3 pharmaceuticals-16-00940-t003:** The irritation score and the type of effect induced by OEO and OEO-PbH, Blank-PbH, and control samples in the HET-CAM assay.

Samples	Irritation Score	Type of Effect
SLS 1%	15.89 ± 0.42	Strong irritant
H_2_O distillate	0	Non-irritant
Blank-PbH	0	Non-irritant
OEO	0	Non-irritant
OEO-PbH	0	Non-irritant

## Data Availability

Not applicable.

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
