# Peer review of "Cutaneous Polymeric-Micelles-Based Hydrogel Containing *Origanum vulgare* L. Essential Oil: In Vitro Release and Permeation, Angiogenesis, and Safety Profile In Ovo"

_pharmaceuticals, 2023, doi:10.3390/ph16070940_

Round 1

Reviewer 1 Report

In the submitted manuscript, the authors conducted research on the in vitro release and permeation, angiogenesis and safety profile in ovo of Cutaneous Polymeric Micelles-Based Hydrogel Containing Origanum vulgare L. Essential Oil. The manuscript is well-written and structurally organized, making it easy to read and understand. Overall, this is an interesting work. However, in my opinion, some aspects need improvement before being considered for publication.

 As a reviewer, I have the following comments:

 The abstract needs to be shortened to convey a more concise message.

  Lines 164, 167 and 169: use "Figure 2b" instead of "Figure 1b".

 The authors should include the major findings of this study in the abstract to make it more attractive and induce readers' interest to read the full text.

  A literature review of previous research on the same topic should be provided, and the novelty of the current manuscript should be highlighted.

 The authors should report all details of materials used (supplier, purity, CAS number, molecular weight, etc.).

  More explanation is needed in Section 4.2.

  References are needed in Sections 4.4. and 4.4.1.

 Please provide more explanation about the in vitro OEO release models used in this study (Section 4.4.3).

 The results of the in vivo assays should be clearly explained, as it is not evident what benefit using this formulation for cutaneous application provides.

 The study should evaluate parameters such as skin irritation, inflammation, systemic toxicity, or any other potential adverse effects.

 Please explain the statistical analysis in more detail. I suggest describing the factors and variables of the study.

 The conclusions should be integrated with more detailed results summarizing the entire study and must reflect the innovation of this study and its perspectives.

Author Response

Dear Dr. Reviewer,

Thank you very much for the time allocated for the article entitled „Cutaneous Polymeric Micelles-Based Hydrogel Containing Origanum vulgare L. Essential Oil: In Vitro Release and Permeation, Angiogenesis and Safety Profile In Ovoas well as for your pertinent observations.

Please see below a point-by-point response to your comments:

In the submitted manuscript, the authors conducted research on the in vitro release and permeation, angiogenesis, and safety profile in ovo of Cutaneous Polymeric Micelles-Based Hydrogel Containing Origanum vulgare L. Essential Oil. The manuscript is well-written and structurally organized, making it easy to read and understand. Overall, this is an interesting work. However, in my opinion, some aspects need improvement before being considered for publication.

As a reviewer, I have the following comments:

Point 1. The abstract needs to be shortened to convey a more concise message.

Response 1: Thank you, we appreciate your indication. We reduced the abstract by eliminationg the less important information.

Point 2. Lines 164, 167 and 169: use „Figure 2b” instead of „Figure 1b.

Response 2: Thank you for this observation. We have corrected this throughout the manuscript.

Point 3. The authors should include the major findings of this study in the abstract to make it more attractive and induce readers' interest to read the full text.

Response 3: Thank you for your valuable suggestion. The most important findings of our study have been added to the abstract.

Point 4. A literature review of previous research on the same topic should be provided, and the novelty of the current manuscript should be highlighted.

Response 4: Thank you, we are grateful for your suggestion. We emphasized the current known data concerning the effects of oregano essential oil in the process of angiogenesis in the Introduction as well as in the Discussion section. To the best of our knowledge, no studies were conducted on polymeric micelles-based hydrogel containing Origanum vulgare L. essential oil regarding the effect on angiogenesis and the possible use in cutaneous applications. Moreover, we pointed the valuable use of the CAM assay for angiogenesis studies and skin intended use of the tested formulation. We highlighted in the Conclusion section the novelty and practical applicability of our research.

Point 5. The authors should report all details of materials used (supplier, purity, CAS number, molecular weight, etc.

Response 5: Thank you for the suggestion. We have added the required details in the manuscript.

Point 6. More explanation is needed in Section 4.2.

Response 6: Thank you for the observation. This work expands our prior investigations conducted by our research group. We completely detailed the preparation of the formulation in our prior study (Bora, L.; Burkard, T.; Juan, M. H. S.; Radeke, H. H.; Muț, A. M.; Vlaia, L. L.; Magyari-Pavel, I. Z.; Diaconeasa, Z.; Socaci, S.; Borcan, F.; et al. Phytochemical Characterization and Biological Evaluation of Origanum Vulgare L. Essential Oil Formulated as Polymeric Micelles Drug Delivery Systems. Pharmaceutics, 2022, 14 (11), 2413. https://doi.org/10.3390/pharmaceutics14112413). Therefore, we added the reference of the abovementioned study in Section 4.2.

Point 7. References are needed in Sections 4.4. and 4.4.1.

Response 7: Thank you for the suggestion. Please note the changes we have made in the Sections 4.4. and 4.4.1., including added references.

Point 8. Please provide more explanation about the in vitro OEO release models used in this study (Section 4.4.3).

Response 8: Thank you for this pertinent observation. Please note that we have explained in more detail the significance of the four mathematical models used to describe the OEO kinetic release from the poloxamer based hydrogel.

Point 9. The results of the in vivo assays should be clearly explained, as it is not evident what benefit using this formulation for cutaneous application provides.

Response 9: Thank you, we are pleased to have your valuable indication. As suggested, we addressed this point, firstly by emphasizing the benefits of employing the chorioallantoic membrane, an in vivo assay, as an excelent accepted alternative method to animal testing, for angiogenesis and skin irritation purposes. Secondly, we explained  the results of the in vivo assay , the reduced number of blood vesssels being indicated using stereomicroscope monitoring, morphometric analisys and histology evaluation, thus pointing to an anti-angiogenic effect and moreover to a possible mechanism involving leptin.

Point 10. The study should evaluate parameters such as skin irritation, inflammation, systemic toxicity, or any other potential adverse effects.

Response 10: Thank you very much for this comment. We have performed in vivo non-invasive measurements in order to observe if any cutaneous toxicity occurs. This assessment is part of our following study–this paper is at present under review–which involves in vivo non-invasive measurements and histological examination of the fibroepithelial polyps.

Point 11. Please explain the statistical analysis in more detail. I suggest describing the factors and variables of the study.

Response 11: We appreciate your observation. We included details for the calculation of an Antiangiogenic index and reguarding the statistically assessed groups.

Point 12. The conclusions should be integrated with more detailed results summarizing the entire study and must reflect the innovation of this study and its perspectives.

Response 12: Thank you for recommendation, we revised the manuscript, adding the explanatory information focused on the most significant results, pointing the novelty and future perspective of the present research. „Finally, this is the first evidence of in vivo antiangiogenic effects of a polymeric micelle-loaded oregano essential oil. Next to the lack of irritation on the CAM tissues, the OEO-PbH formulation proves good bioavailability and safety use intended for cutaneous applications. Moreover, results pointed to mechanistic insights of the OEO-PhB formulation, involving leptin as a possible target. However, further research is necessary to substantiate these findings, investigate the exact mechanism of action, and optimize the formulation for maximal therapeutic efficiency and patient comfort in clinical settings.”

Thank you once again for your suggestions. We hope that we have improved the article’s scientific quality as per your consideration.

Best regards,
Larisa Bora, PhD student

Department of Pharmacognosy

Victor Babes University of Medicine and Pharmacy Timisoara, Romania.

Reviewer 2 Report

This is a well-written paper by Avram and colleagues evaluating the functionality of polymeric micelles-based hydrogel (PbH) for delivery of oregano essential oil (OEO) to skin lesions.

Author documented that OEO-PbH formulation produced a sustained prolonged release of OEO through the hydrophilic membrane and respectively pig ear skin; by using CAM method authors showed lack of irritability induced by OEO-PbH and identified an effect of limiting the number of capillaries - angiogenesis inhibition by 30-40%.

Authors concluded, that their study provides a potential foundation for further exploration into the clinical utility of OEO-PbH in the management of cutaneous fibroepithelial polyps. OEO-PbH's biocompatibility, lack of irritability, and limiting effect on angiogenesis suggest a potential for its therapeutic application, and its sustained release could enhance patient compliance and improve therapeutic outcomes.

This is an interesting study and clinically valuable. The study was well performed and authors extensively discussed the results. The ideas and conclusions presented in this manuscript may attract attention of some groups of researchers, especially those searching for new transdermal drug application method.

Minor comments:

Comment 1. I am not expert in CAM method however I would expect that assessement of OEO-PbH activity on chick chorioallantoic membrane is related to the same observation field before and after sample application, see Figure 3. The in ovo stereomicroscopic images of the HET-CAM assay illustrating effects initially (t0) 195 and 300 s (t5) after contact with the samples

or during time-course experiments, see Figure 4. Effects of OEO-PbH on normal angiogenesis on the CAM.

In both Figures (3 and 4) showed images do not represent the same microscopic fields thus it is hard to compare them and indicate subtle changes/effects in CAM morphology upon treatments (except for SDS). Please comment on these issues.

Comment 2. Sensitization to herbs commonly used in cuisine and cosmetics, although rare, is being increasingly reported. The herbs and spice sensitizations are often underrated or misdiagnosed during routine allergy evaluation. Importantly, positive skin prick test (SPT) with herbs may be indicatory for a patient to restrain potential health-threatening episodes through control of spice consumption/contact. According to Wagner et al. (Int J Environ Res Public Health. 2022 Dec 20;20(1):33. doi: 10.3390/ijerph20010033) children with suspected allergy to aeroallergens (mostly birch, mugwort and grass pollen) are prone to develope skin reactions upon skin prick challange (SPT) with oregano extract. The susceptibility of the adult and pediatric patients to developing SPT reactions were accordingly 24% and 30%. Please comment on these issues in context of  OEO use in skin application.

Taken together, this manuscript by Avram and colleagues represents a worthwhile contribution to the pharmaceutical research. I recommend the manuscript for further publication process.

A few typographical mistakes

Author Response

Dear Dr. Reviewer,

Thank you very much for the time allocated for the article entitled „Cutaneous Polymeric Micelles-Based Hydrogel Containing Origanum vulgare L. Essential Oil: In Vitro Release and Permeation, Angiogenesis and Safety Profile In Ovoas well as for your pertinent observations.

Please see below a point-by-point response to your comments:

This is a well-written paper by Avram and colleagues evaluating the functionality of polymeric micelles-based hydrogel (PbH) for delivery of oregano essential oil (OEO) to skin lesions.

Author documented that OEO-PbH formulation produced a sustained prolonged release of OEO through the hydrophilic membrane and respectively pig ear skin; by using CAM method authors showed lack of irritability induced by OEO-PbH and identified an effect of limiting the number of capillaries - angiogenesis inhibition by 30-40%.    

Authors concluded, that their study provides a potential foundation for further exploration into the clinical utility of OEO-PbH in the management of cutaneous fibroepithelial polyps. OEO-PbH's biocompatibility, lack of irritability, and limiting effect on angiogenesis suggest a potential for its therapeutic application, and its sustained release could enhance patient compliance and improve therapeutic outcomes.

This is an interesting study and clinically valuable. The study was well performed and authors extensively discussed the results. The ideas and conclusions presented in this manuscript may attract attention of some groups of researchers, especially those searching for new transdermal drug application method.

Minor comments:

Comment 1. I am not expert in CAM method however I would expect that assessement of OEO-PbH activity on chick chorioallantoic membrane is related to the same observation field before and after sample application, see Figure 3. The in ovo stereomicroscopic images of the HET-CAM assay illustrating effects initially (t0)         and 300 s (t5) after contact with the samples

or during time-course experiments, see Figure 4. Effects of OEO-PbH on normal angiogenesis on the CAM.

In both Figures (3 and 4) showed images do not represent the same microscopic fields thus it is hard to compare them and indicate subtle changes/effects in CAM morphology upon treatments (except for SDS). Please comment on these issues.

Response 1: Thank you for your pertinent observation. We addressed your inquiries while adding more clarity to the specific results section in the manuscript. 

The experimental procedure of the HET-CAM assay involves the continuous observation of the whole area of application with higher magnification for 300 seconds, capturing the most representative images for the events of interest (i.e., coagulation, hemorrhage, lysis).  In the entire treated area, no signs of any effect were observed.

Regarding the angiogenesis evaluation, the CAM experimental setting involves daily application of samples and careful observation of the treated area, at a higher magnification, by means of stereomicroscopy. The image of the field may seem different due to the rapid growth of blood vessels and the development of vessel architecture, which are typical for the experimental time frame.  The higher magnification also allowed counting the number of vessels intersecting the ring, and consequently the calculation of the angiogenic index.

Comment 2. Sensitization to herbs commonly used in cuisine and cosmetics, although rare, is being increasingly reported. The herbs and spice sensitizations are often underrated or misdiagnosed during routine allergy evaluation. Importantly, positive skin prick test (SPT) with herbs may be indicatory for a patient to restrain potential health-threatening episodes through control of spice consumption/contact. According to Wagner et al. (Int J Environ Res Public Health. 2022 Dec 20;20(1):33. doi: 10.3390/ijerph20010033) children with suspected allergy to aeroallergens (mostly birch, mugwort and grass pollen) are prone to develope skin reactions upon skin prick challange (SPT) with oregano extract. The susceptibility of the adult and pediatric patients to developing SPT reactions were accordingly 24% and 30%. Please comment on these issues in context of  OEO use in skin application.

Taken together, this manuscript by Avram and colleagues represents a worthwhile contribution to the pharmaceutical research. I recommend the manuscript for further publication process.

Response 2: Thank you very much for this observation and for raising this aspect. The present study represents a preliminary study between in vitro determinations already published (Bora, L.; Burkard, T.; Juan, M. H. S.; Radeke, H. H.; Muț, A. M.; Vlaia, L. L.; Magyari-Pavel, I. Z.; Diaconeasa, Z.; Socaci, S.; Borcan, F.; et al. Phytochemical Characterization and Biological Evaluation of Origanum Vulgare L. Essential Oil Formulated as Polymeric Micelles Drug Delivery Systems. Pharmaceutics, 2022, 14 (11), 2413. https://doi.org/10.3390/pharmaceutics14112413) and clinical evaluation on human subjects (paper under review). The present study is a preliminary evaluation in order to outline the mechanisms of action of the hydrogel at blood vessels level. We also conducted a comprehensive clinical study by means of non-invasive measurements, clinical evaluation, and histological examination of fibroepithelial polyps after local treatment with OEO-PbH – this paper is at present under review. We also conducted preliminary studies in order to observe the safety profile (patch test).

It is important to mention that Wagner ̀s study evaluated an oregano extract, not the volatile oil. Moreover, in order to reduce the local toxicity, we incorporated oregano volatile oil into polymeric micelles.    

Thank you once again for your suggestions. We hope that we have improved the scientific quality of the article as per your consideration.

Best regards,
Larisa Bora, PhD student

Department of Pharmacognosy

Victor Babes University of Medicine and Pharmacy Timisoara, Romania.
